# Extramedullary Hematopoiesis of the Liver and Spleen

**DOI:** 10.3390/jcm10245831

**Published:** 2021-12-13

**Authors:** Diana Cenariu, Sabina Iluta, Alina-Andreea Zimta, Bobe Petrushev, Liren Qian, Noemi Dirzu, Ciprian Tomuleasa, Horia Bumbea, Florin Zaharie

**Affiliations:** 1Medfuture Research Center for Advanced Medicine, School of Medicine, Iuliu Hatieganu University of Medicine and Pharmacy, 400337 Cluj-Napoca, Romania; diacenariu@gmail.com (D.C.); zimta.alina.andreea@gmail.com (A.-A.Z.); bobe.petrushev@gmail.com (B.P.); noemidirzu@gmail.com (N.D.); 2Department of Hematology, Iuliu Hatieganu University of Medicine and Pharmacy, 400124 Cluj-Napoca, Romania; sabina.iluta@umfcluj.ro; 3Department of Hematology, Municipal County Hospital, 400111 Oradea, Romania; 4Department of Pathology, Octavian Fodor Regional Institute of Gastroenterology and Hepatology, 400124 Cluj-Napoca, Romania; 5Department of Hematology, 5th Medical Center of the People’s Liberation Army General Hospital, Beijing 100037, China; qlr2007@126.com; 6Department of Hematology, Ion Chiricuta Clinical Cancer Center, 400004 Cluj-Napoca, Romania; 7Department of Hematology, Carol Davila University of Medicine and Pharmacy, 400004 Bucharest, Romania; 8Department of Hematology, University Emergency Hospital, 400004 Bucharest, Romania; 9Department of Surgery, Iuliu Hatieganu University of Medicine and Pharmacy, 400124 Cluj-Napoca, Romania; zaharie.vasile@umfcluj.ro

**Keywords:** extramedullary hematopoiesis, spleen and liver, embryology, chronic myeloproliferative neoplasms

## Abstract

Hematopoiesis is the formation of blood cellular components and, consequently, immune cells. In a more complete definition, this process refers to the formation, growth, maturation, and specialization of blood cells, from the hematopoietic stem cell, through the hematopoietic progenitor cells, to the s pecialized blood cells. This process is tightly regulated by several elements of the bone marrow microenvironment, such as growth factors, transcription factors, and cytokines. During embryonic and fetal development, hematopoiesis takes place in different organs: the yolk sac, the aorta–gonad mesonephros region, the lymph nodes, and not lastly, the fetal liver and the spleen. In the current review, we describe extramedullary hematopoiesis of the spleen and liver, with an emphasis on myeloproliferative conditions.

## 1. Introduction

Hematopoiesis (HMP) is the formation of blood cellular components and, consequently, immune cells. In a more complete definition, this process refers to the formation, growth, maturation, and specialization of blood cells, from the hematopoietic stem cell (HSC), through the hematopoietic progenitor cells (HPCs), to the specialized blood cells [1]. This process is tightly regulated by several elements of the bone marrow (BM) microenvironment, such as growth factors, transcription factors, and cytokines.

During embryonic and fetal development, hematopoiesis takes place in different organs: the yolk sac, the aorta–gonad mesonephros region, the lymph nodes, and not lastly, the fetal liver and the spleen [2]. By the end of the second trimester, hematopoiesis is settled in the bone marrow, where it will continue throughout life. In certain pathological circumstances, mainly as a compensatory mechanism for the destruction of the proper function of the BM, genesis of blood cells moves to other organs, mostly the ones involved in fetal hematopoiesis. This process, called extramedullary hematopoiesis (EMH), is an important, yet little understood, feature of infection, certain hematological diseases, and gastrointestinal or other solid neoplasms. It poses great difficulties in the differential diagnosis of hepato/splenomegaly. It also can cause complications at the gastrointestinal level, such as bleeding and polyps [3,4].

Through this review, the authors attempt to bring light to the involvement of extramedullary hematopoiesis in the complex diagnosis of hepato-splenomegaly in hematological diseases, with a special focus on myeloproliferative neoplasia and fetal development of hematopoiesis.

There are two different stages of fetal hematopoiesis: primitive and definitive HMP. Primitive HMP takes place during the early fetal development in the yolk sac [5]. The liver and heart are the first internal organs that develop; their buddings are rapidly seen in early embryonic development. Before becoming a part of the gastrointestinal tract, the liver is part of the cardiovascular system. During the gastrulation of the blastocyte, the clumps of cells are divided into three germinal layers: exoderm, mesoderm, and endoderm. The endoderm is then divided into the foregut, midgut, and hindgut, in the transverse septum. This forms a junction between the ectoderm of the amnion and the endoderm of the yolk sac, externally, and a junction between the foregut and midgut, internally. This junction has a mesenchymal structure, and it is the origin of the liver and the major blood vessels. The liver forms at day 8 of embryogenesis. During this period, this organ is part of the cardiovascular system through its hematopoietic function, and by interconnecting the early placental vessels with the heart. In early hepatic hematopoiesis, the first cells that are generated are the undifferentiated mononuclear cells (future hematopoietic stem cells). The primitive erythroid cells are the most abundant cell lines found at this stage [6], but the presence of mast cells, and natural killer and innate lymphoid cell precursors in the yolk sac, were also detected [5]. The erythroblasts or megaloblasts produced at this stage are not pluripotent or self-renewing, because the main interest of the body, at this point, is to ensure rapid embryonic growth and development.

On the other hand, definitive HMP will converge towards HSC and HPC production first in the aorta–gonad mesonephros (AGM) region, later in the fetal liver, until the 4th–5th months of pregnancy. HSCs will migrate to the bone marrow (around week 11, post-conception), to the spleen and lymph nodes, as well. At this gestational time, the microenvironment is favorable to self-renewal and differentiation of HSCs/HPCs. As the fetus develops, the number of generated stem cells decreases, and the number of erythroid cells increases. In later stages, there are no immature erythroid cells to be detected in the hepatic area, whereas they are found in the extravascular space [3]. Thus, during embryogenesis, the hematopoiesis function of the liver is especially related to its erythroid function. This function is taken from the yolk sac [4]. During this transition, the primitive blasts from the yolk sac migrate to the liver. The yolk sac has a more immature, less differentiated type of erythropoiesis, where the starting cells are derived from the mesodermal cells, whereas in the liver, the cells of origin are hematopoietic stem cells [5]. Besides these, the liver is colonized with HSCs from a hemogenic endothelium on the ventral wall of the aorta [6]. In the liver, the cells of origin have a lesser difference of maturation between the nucleus and cytoplasm, and a smaller number of polyribosomes [3]. The hepatic erythrocytes rapidly differentiate and take the place of the circulating HSCs found in early embryonic development [6]. In general, hepatic erythropoiesis has many similarities to bone marrow erythropoiesis. They both start from hematopoietic stem cells, which mature into erythroid progenitors, such as proerythroblasts, basophilic erythroblasts, polychromatophilic erythroblasts, and orthochromatic erythroblasts. These cells possess colony-formation capacity in vitro. One orthochromatic erythroblast undergoes an asymmetric division and divides in two cells: one containing the nucleus and a small cytoplasm (pyrenocyte), and one enucleated cell called a reticulocyte, which later matures into a red blood cell [5]. Hepatic hematopoiesis has a very poor representation of the microtubules typically found in myeloid erythroblasts [3]. Hepatic hematopoiesis occurs very early in embryonic development, in the 7th week of gestation. After the 20th week, the spleen fulfills the same function of hematopoiesis along with the liver. The liver continues to function as a hematopoietic organ until the third trimester, when the bones are fully ossified and the bone marrow takes over [7]. The erythrocytes at this point contain fetal hemoglobin (HbF), with two γ-globin chains and two alpha-globin chains, until the 32nd week of gestation, when the adult hemoglobin (HbA) replaces the HbF [7].

A comprehensive study, with detailed information on the origin of blood and immune development in fetal liver, yolk sac, skin, and kidney; offering insights into fetal erythropoiesis, seeding of mast cells, natural killer and lymphoid cells from the yolk sac, the potential dual myeloid and lymphoid origin of plasmacytoid dendritic cells, modulation of HSCs/HPCs, intrinsic differentiation potential, and more data, is presented in an exquisite paper by Popescu et al., publishing an atlas of human early cell growth, differentiation, maturation, and specialization. The team conducted a complex study by molecular mapping using single-cell transcriptomics, in order to provide a deep understanding of human fetal-cell behavior between the 7th–17th post-conception weeks, when the liver is the major site for fetal hematopoiesis erythroid cells, megakaryocytes, and when mast cells are present in the non-lymphoid tissues (NLT), but HSC and multipotent progenitor cells (MPPs) are absent. NLTs are colonized by macrophages, monocytes, myeloid progenitors, and dendritic cells, as soon as one week after conception [5].

Self-renewing HSCs populate the fetal liver from 6 weeks to 22 weeks of gestation, and they encounter a more favorable terrain for supporting their growth and expansion. The explanation comes from the higher number of regulators of the Wnt signaling pathway present in the fetal liver stroma, as opposed to the BM microenvironment, where the Notch signaling pathway expression is higher. HSCs in adulthood are residents of the bone marrow stromal microenvironment, mostly quiescent and with a limited self-renewal [7]. Experimental data on human cells derived from a 15- to 21-week fetal liver, validated the above-mentioned observations that these stromal cells have a larger proliferative potential, 10-fold growth in cell number, and a longer lifespan than bone marrow stromal cells. Wnt signaling pathway genes were up regulated in the fetal liver, explaining the higher self-renewal rate and differentiation of HSCs [8].

The fetal BM becomes the leading site of hematopoiesis after 20 post-conception weeks in humans. The BM niche with the cartilaginous bone, invaded by blood vessels and bone ossification, represents the perfect environment for HSC and HPC spreading and infiltration. Once the BM has fully matured, HMP will predominantly take place in the BM niche, and the embryonic period of hematopoiesis will be over [9].

## 2. The Bone Marrow Hematopoietic Niche

HSCs are non-specialized cells from which all blood cells derive. They are maintained throughout the whole lifespan in order to supply the organism with new blood cells after physiological and pathological destruction, and after all kinds of assaults. By their self-renewal capacity, they can be divided into long-term HSCs and short-term HSCs [10]. They undergo an asymmetrical division, which allows them to maintain their number and differentiate at the same time, by giving birth to a new HSC and a cell committed to differentiation—the hematopoietic progenitor cell (HPC). In physiological circumstances, HSCs and HPCs are present mainly in the bone marrow and in the peripheral blood [1]. They receive signals and commands on their future differentiation path from a unique microenvironment, called the hematopoietic niche, found near the sinusoid blood vessels [11]. Progenitor cells with different lineages occupy distinct niches in the bone marrow, together with cells that secrete mediators, which will dictate their path of differentiation into mature cells, such as erythrocytes, lymphocytes, granulocytes, and megakaryocytes belonging to the myeloid or lymphoid line. They can also maintain a stable pool of HPCs by self-renewal, ensuring a tissue homeostasis [12], but at a much lower level than HSCs [13].

HSCs can be detected in blood, using flow cytometry by identifying the presence or the absence of certain surface antigens [14]. CD34 is considered a universal HSC surface marker in humans [15]. Diagnosis from solid tissues, however, requires more invasive techniques, such as biopsy, fine-needle aspirate, or surgical removal of the site [16]. HPCs have an endothelial origin, due to their onset from the aortic endothelial layer. These cells exhibit endothelial markers, which is proof that the endothelial-to-hematopoietic transition is a common process for blood formation, regardless of the developmental stage. It is difficult to identify the specific markers that lead to the formation and expansion of HSCs, and to separate them from the non-self-renewing progenitors. Once this dilemma is solved, many questions will find answers, and new clinical treatments for hematological diseases will emerge [9].

## 3. Extramedullary Hematopoiesis

In certain pathological states, HSCs and HPCs can be released in large numbers into the bloodstream, and they migrate to other organs. Extramedullary hematopoiesis (EMH) is the formation and activation of blood cells outside the bone marrow (BM), as a response to hematopoietic stress caused by microbial infections and certain diseases, such as myeloproliferative neoplasms (MPN), lymphomas, and leukemias, when the proper functioning of the marrow is deteriorated (Figure 1). The HSCs in the bone marrow are the sources of immature progenitor cells of myeloid and lymphoid origin. Once hematopoietic differentiation is disrupted, there is an increased chance of developing certain blood cancers: leukemia, lymphoma, or myeloma [17].

Both the spleen and the liver are major constituents of the portal circulation, which comprises important reticuloendothelial structures that take part in substance exchange and cellular migration. They are key players in immune homeostasis [18]. The liver, under normal conditions, has a secretory function, but under certain circumstances, it works to compensate for the bone marrow’s deficiency and, together with the spleen, attempts to remedy the lack of functional blood cells. In the liver, they appear as heterogenous masses, which may be confused with metastatic lesions, but at biopsy, EHP is present with dilated and congested sinuses and precursor cells [19]. The spleen is an organ with high metabolic activity, positioned in the upper-left abdomen, underneath the diaphragm, close to the stomach and the pancreas, and is protected by the ribcage. Its role is to recycle blood particles and iron, and to filter atypical blood cells; to phagocyte aged erythrocytes, damaged platelets, and apoptotic cells; to produce and shelter antibodies; to sustain the immune system in fighting infections by differentiation; and to activate T and B cells [19,20]. The white pulp is responsible for the immune response to infection by activation of T- and B-lymphocytes, which eventually leads to IgM and IgG antibody production. The red pulp extends up to 80% of the spleen parenchyma, mainly in the cords tissue (rich in macrophages), and the venous sinuses. Macrophages phagocyte all the abnormal cells in the cords and opsonize bacteria. The sensitivity of asplenic or splenectomized mice to massive bacterial infections is caused by the lack of IgM memory B cells [21]. The spleen is regarded as the main site of EMH in different hematopoietic disorders: primary myelofibrosis (PMF) [22,23], acute myeloid leukemia (AML), hemolytic anemia, thalassemia, non-Hodgkin’s lymphoma (NHL), immune thrombocytopenic purpura (ITP) [24], as well as spondylarthritis [25], atherosclerosis [26], and osteopetrosis (marble bone disease) [8]. The numerous splenic EMH diseases sustain the conclusion that the spleen is the main EMH site, but it is functionally and mechanistically different from the BM regarding blood cell formation [9]. Table 1 details the studies found in the literature regarding EMH in hematological and non-hematological diseases.

As a result of severe infection and inflammation, the spleen and the liver will regain their embryologic role in hematopoiesis. For example, murine hematopoietic stem cell transplantation redirects megakaryopoiesis to the spleen for an increased megakaryocyte maturation [27]. Acute inflammation triggers rapid maturation of megakaryocyte progenitors, and increased platelet production by direct implication of both mTOR and STAT1 signaling pathways in megakaryopoiesis [28]. Reduced GATA-1 expression activates in mice a disorder similar to human primary myelofibrosis, characterized by myelofibrosis, extramedullary hematopoiesis, and anemia. STAT1 is an essential effector of GATA-1 during normal megakaryocyte production, and the activation of this pathway eventually leads to thrombocytosis in mouse models [27]. Activation of Tlx1 transcription factor determines the mobilization and proliferation of HSCs and HPCs, leading to the development of a niche-induced extramedullary hematopoiesis in the spleen [29]. Splenic myelopoiesis occurs in cancer, as a supplementary production of myeloid cells, which are important regulators of disease progression in the tumor microenvironment [19].

**Table 1 jcm-10-05831-t001:** Extramedullary hematopoiesis in clinical studies and experimental settings.

Pathology	Sites of EMH	Lineage	Number of Cases	Human/Animal Studies	References
Primary myelofibrosis	Spleen liver		5 patients	Human	[22]
Secondary myelofibrosis after essential thrombocythemia	Spleen		1 patient	Human	[22]
Primary myelofibrosis	Spleen	Malignant primitive HPCs (PB CD34+)	8 patients	Human	[23]
Allograft liver transplant	Liver		27 patients		[30]
Acute promyelocytic leukemia (APL)	Central nervous system, skin, lung,	Leukemia cells	21 patients	Human	[31]
Chronic myelogenous leukemia	Stomach		Case reports	Human	[3,4,32]
Thalassemia	N/A		10 patients	Human	[33]
Agnogenic myeloid metaplasia	Lung		2	Human	[34]
Primary myelofibrosis (MMM)	Spleen	Granulocyte infiltration	213 patients (MMM)	Human	[35]
Myelofibrosis with myeloid metaplasia (MMM) with thalidomide therapy	Liver/spleen Pericardium	Red blood cells Granulocytic precursors	1 of 7 patients	Human	[36]
Chronic myeloproliferative diseases	Spleen	JAK2(V617F)	15 patients 47 patients	Human Human	[37] [38]
Experimental spondylarthritis	Spleen Inflamed joints	Myelopoiesis	39 mice (13 mice/group)	Animal/experimental	[25]
Inflammation	Spleen, liver, other sites	Megakaryopoiesis	N/A	Animal	[28]
Experimental mutations in the SETD1B gene;	Spleen	Myeloid and lymphoid	N/A	Animal	[39]
Acute myeloid leukemia	Spleen		N/A	Animal	[40]
Experimental overexpression of Tlx1	Spleen		N/A	Animal	[29]
Experimental Dnmt3a loss of function	Spleen, liver	HSC	18 mice	Animal	[41]

Scientists explain the effects of epigenetic modifiers on normal and malignant hematopoiesis [42,43,44,45]. The most frequent is CpG (cytosine–guanine nucleotide) methylation carried out by the DNA methyltransferases family. In the recent years, DNMT3A (DNA (cytosine-5-)-methyltransferase 3 alpha) and TET2 gene mutations affecting DNA methylation, are detected by genome sequencing in many myeloproliferative neoplasms [43,46,47]. DNMT3A mutations at codon R882H may contribute to therapy resistance, apoptosis, and DNA repair blocking [48]. A complete understanding of the epigenetic phenomenon in hematopoiesis is yet to be discovered. Hopefully, epigenome editing techniques will unravel DNA methylation of gene expression and histone modifications that regulate gene transcription and chromatin structure [49].

## 4. Pathological Hematopoiesis

There are situations in which the normal process of hematopoiesis is disrupted, and abnormal proliferation of one or more myeloid lines occurs, which is a pathology known as a myeloproliferative neoplasm. William Dameshek observed and described the concept of myeloproliferative disorders in 1951, a concept that has now been reformed by the World Health Organization (WHO) into myeloproliferative neoplasms [50]. This heterogeneous group of disorders has several common features, such as overproduction or proliferation of one or more blood elements having as their dominant element a transformed clone, hypercellular marrow or bone marrow fibrosis, cytogenetic abnormalities, hemostasis disorders, such as thrombosis or hemorrhage, or EMH—mainly in the spleen or the liver—transformation into an acute condition, such as acute leukemia [50].

The WHO classification includes four major classical myeloproliferative neoplasms: polycythemia vera (PV), essential thrombocythemia (ET), chronic myeloid leukemia (CML), and primary myelofibrosis (PMF). It also describes chronic neutrophilic leukemia, chronic eosinophilic leukemia, and unclassifiable MPNs that are rare, with their incidence and prevalence being unknown, due to their rarity [51]. Classical myeloproliferative neoplasms can be divided into BCR-ABL positive (Philadelphia chromosome) and BCR-ABL negative. Of these, only chronic myeloid leukemia is BCR-ABL positive, the rest are negative [50].

Like most malignant hematological diseases, the exact etiology of myeloproliferative neoplasms is unknown. However, it has been observed that patients who test positive for the JAK2 genetic mutation have a higher risk for developing MPNs [52,53]. A low dose of ionizing radiation has been associated with an increased risk of developing MPNs, due to somatic mutations in the JAK2 gene (JAK2 V617F variant) [54]. New data point out that there is an association between environmental exposure and somatic mutations in environment-sensitive genes, such as homozygosity of CYP1A1 rs4646903, CYP1A2, EPHX1 rs2234922, and Tp53 alleles [55].

Molecular studies in recent years, and a better understanding of the molecular pathogenesis of these diseases, have led to a change in their clinical approach. Mutations within MPNs are called restricted. They are not found in other myeloid malignancies, while unrestricted mutations can be found in other myeloid malignant hematological diseases. The presence of these mutations in hematopoietic stem cells leads to the clonal expansion of one or more cell lines developing MPNs [56].

The main mutation present in CML is the Philadelphia chromosome. This chromosome arises from the reciprocal translocation of chromosomes 22 and 9, shortening the length of chromosome 22. Translocation consists of the fusion of the cluster breakpoint gene (BCR) on chromosome 22, together with oncogenes of the Abelson murine leukemia virus (ABL) on the long arm of chromosome 9, thus forming the BCR-ABL fusion gene [57,58].

Fusion of the ABL oncogene with the BCR gene leads to increased ABL tyrosine kinase activity, by encoding a chimeric protein in the BCR-ABL fusion gene. This fusion gene can be tested positive in acute lymphoblastic leukemia (ALL) also [59].

On the other hand, negative BCR-ABL MPNs have as restricted mutations Janus kinase 2 (JAK2) [60], the myeloproliferative leukemia virus proto-oncogene (MPL), calreticulin (CALR) [61], and the colony-stimulating factor 3 receptor (CSF3R), encoding different tyrosine kinases [62].

The pathogenesis of MNPs on the JAK2 pathway consists of the activity of this protein kinase that phosphorylates the signal transducer and activator of transcription (STAT). The JAK2 mutation is a somatic mutation, and occurs by substituting valine with phenylalanine at codon 617 in the pseudokinase domain (JAK2 V617F). It is present in approximately 70% of MPNs, with a frequency of approximately 95% in patients with PV, 50–70% in patients with ET, and 40-50% in patients with PMF [63].

Regarding the MPL gene, it encodes the thrombopoietin receptor (TPO), which regulates megakaryopoiesis through a JAK-STAT pathway [64]. Among the MPL mutations, the most common are MPL W515L and MPL W515K [65]. Abnormal proliferation of hematopoietic stem cells results from spontaneous activation of the JAK-STAT pathway by MPL mutations [66,67]. Determining this gene is important because it can differentiate patients with a negative JAK2V617F mutation and those with a positive BCR-ABL mutation. A particularity of patients who are MPL positive is the increased risk of thrombotic complications and the association with low hemoglobin levels, poor bone marrow cellularity, and elevated serum erythropoietin levels. The CALR gene is located on chromosome 19, exon 9. This gene is responsible for encoding calreticulin, a calcium-binding protein in the endoplasmic reticulum that regulates cell proliferation, differentiation, and apoptosis. This gene may occur in JAK2 and MPL negative patients, being first observed in 2013. The CSF3R gene can be detected in chronic neutrophilic leukemia, which is also a WHO diagnostic criterion for this disease [68].

The characteristics of the splenic stem cells of PMF patients seem to differ from those of BM origin. A significant number of self-renewing HSCs belong to B- and T- cell lineages in PMF spleens. Taking into consideration that PMF originates from a cancerous stem cell, the above-mentioned PMF-SCs colonize extramedullary sites (spleen and liver) that confer the nurturing niche for preserving their functional properties [23]. ^111^InCl3-transferrin scintigraphy detected the intensity of splenic EMH in PMF patients, as a result of high inflammatory processes [22].

CML is characterized by the presence of three phases of evolution: chronic, accelerated, and blastic. In the chronic phase, the hypercellular marrow is described with a left deviation of the leukocyte formula and myeloblasts up to 10%. The accelerated phase shows an increase in the number of myeloblasts up to 19% and basophilia 20%; chromosomal abnormalities may be present in this phase. The blast phase represents the transformation into acute myeloid leukemia, with a percentage of myeloblasts over 20%. Peripheral blood is positive for leukocytosis, with slight elevation of eosinophils, basophils, and monocytes.

Polycythemia vera is characterized by trilinear proliferation, with increased hemoglobin values, over 16.5 g/dL in men and over 16.0 g/dL in women. The transformation of polycythemia vera into myelofibrosis may reveal leukoerythroblastosis, poikilocytosis, and tear-shaped erythrocytes [69]. In essential thrombocythemia, thrombocytosis is significant, without a secondary cause, and is associated with platelet anisocytosis. Myelofibrosis after thrombocythemia describes changes similar to those in polycythemia, and in the bone marrow, there is mild hypercellularity without macrocytosis and dysgranulopoiesis. Primary myelofibrosis describes a leukoerythroblastic picture, with immature granulocytes and nucleated erythrocytes. The medullary aspirate is dry, and an osteo-medullary biopsy is required, which describes different degrees of fibrosis and atypia of megakaryocytes.

The symptoms of MPNs are varied and relatively similar. Patients describe clinical manifestations such as fatigue, anorexia, sweating, weight loss, abdominal discomfort or even increased abdominal pain, bleeding, headache, and dizziness. Clinically, hepatosplenomegaly is the main characteristic of these patients. The spleen and the liver are palpable, pain may be present in the upper-left quadrant of the abdomen, due to splenic infarction. Due to ineffective coagulation, patients tend to bruise easily or exhibit signs of thrombosis [70]. Polycythemia vera is characterized by hyper-viscosity, and may be associated with symptoms, such as hypoxia or thrombosis [71]. In essential thrombocythemia, the symptomatology varies from asymptomatic to manifestations related to microvascular occlusions of the distal vessels, causing erythromelalgia or gangrene. Splenomegaly is more common than hepatomegaly in thrombocythemia. In primary myelofibrosis, signs of medullary insufficiency may occur, such as bleeding, petechiae or bruising, or infections due to neutropenia.

The diagnosis of MPNs is made through a series of evaluations. These include complete blood count (elevated levels of one or more cell lines), blood smear (leukoerythroblastosis and giant platelets), electrolytes, leukocyte alkaline phosphatase, uric acid, lactate dehydrogenase, erythropoietin level, von Willebrand factor determination in patients with platelets over 1 million/µL, bone marrow aspiration and biopsy (hypercellularity), cytogenetic analysis for Philadelphia chromosome, and molecular testing for JAK2, MPL, CALR, and CSF3R [72].

A common feature in terms of physical examination related to MPNs is hepatosplenomegaly. As modern drugs attack different genetic targets, it is of paramount importance to establish a correct etiological diagnosis of the different MPNs and other hematological malignancies, such as acute leukemias or non-Hodgkin’s lymphoma [73]. It is also important to distinguish between a malignant or benign cause of hepatosplenomegaly. It is important to accurately and thoroughly establish the differential diagnosis of the hepatosplenomegaly from MPN against hepatic diseases (cirrhosis and hepatitis), acute or chronic infections (mononucleosis, tuberculosis, abscess, and HIV), venous thrombosis in the portal vein, splenic sequestration from hemolytic anemias, thalassemia or pediatric sickle cell, focal lesions in the spleen (hemangiomas, abscesses, metastases, or cysts), connective tissue diseases (systemic lupus erythematosus and rheumatoid arthritis), and infiltrative diseases (amyloidosis and sarcoidosis) [74].

## 5. Conclusions

There are two different stages of fetal hematopoiesis: primitive and definitive HMP. In the current review, we described both primitive hematopoiesis of the liver and spleen, as well as secondary hematopoiesis of these organs, of special interest in chronic myeloproliferative neoplasms.

## Figures and Tables

**Figure 1 jcm-10-05831-f001:**
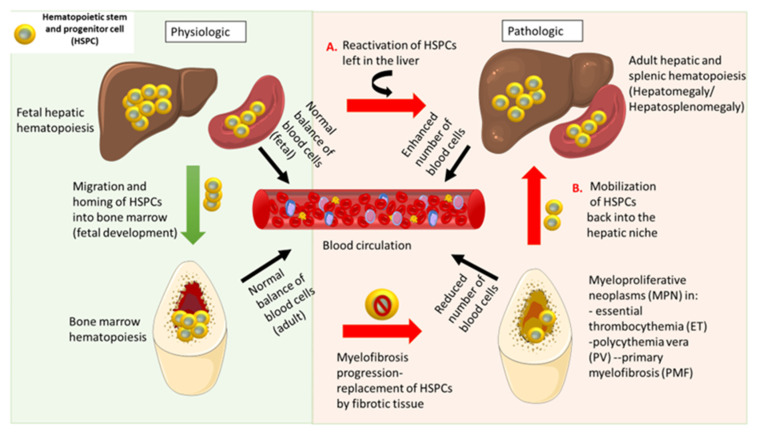
Physiological versus pathological hematopoiesis.

## Data Availability

Upon request, the authors are willing to provide any required information.

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
