# Peer review of "Extramedullary Hematopoiesis of the Liver and Spleen"

_jcm, 2021, doi:10.3390/jcm10245831_

Round 1

Reviewer 1 Report

The authors provide a review of extramedullar hematopoiesis in pathological conditions, with particular regards to myeloproliferative disorders . 

Global comments:

The review globally reads quite well but to my opinion it would benefit from a better harmonisation among the different sessions. It looks like the different sessions have been written separately by different authors, while a global harmonisation is somehow missing.

The different sessions are not homogeneously developed; for exemple the fetal hematopoiesis session is quite detailed unlike others such as extramedullary hematopoiesis although this is the core of the manuscript.

Also, in the "extramedullary hematopoietic" session the splenic and liver hematopoiesis are mentioned, a description of the splenic hematopoiesis is provided but this is missing for the liver hematopoiesis.

Finally, the authors could mention the additional contribution of their manuscript to the existing literature.

Specific comments:

  • a lot of abbreviations are used in the manuscript, some of them are not commonly used, which makes the reading somehow confusing. I would suggest to reduce those abbreviations that are not necessary; also, for example hemoglobin is more commonly abbreviated as "HbA, HbF" instead of "HgA, HgF",  I therefore suggest to adopt commonly used abbreviations for a easier reading.
  • a table is shown in the manuscript, but it is not clear its role and contribution to the manuscript. also, this table is not clearly mentioned in the text.
  • line 106 : MPP to be defined
  • paragraph lines 142 to 150 the link between the different sentences of this paragraph is not clear , please revise.
  • line 159: HSC are again defined but this is done since the beginning of the manuscript
  • Table: line "Inflammation": in the "number of cases" column it is noted "in vitro, in vivo", please revise/complete.
  • table titer and number are missing.

Author Response

Reviewer 1

The authors provide a review of extramedullar hematopoiesis in pathological conditions, with particular regards to myeloproliferative disorders .

Global comments:

The review globally reads quite well but to my opinion it would benefit from a better harmonisation among the different sessions. It looks like the different sessions have been written separately by different authors, while a global harmonisation is somehow missing.

The different sessions are not homogeneously developed; for exemple the fetal hematopoiesis session is quite detailed unlike others such as extramedullary hematopoiesis although this is the core of the manuscript.

Also, in the "extramedullary hematopoietic" session the splenic and liver hematopoiesis are mentioned, a description of the splenic hematopoiesis is provided but this is missing for the liver hematopoiesis.

Thank you for the very constructive suggestion. All changes have been made in the revised manuscript, in red. We added to the manuscript a section detailing extramedullary hematopoiesis in the liver.

Finally, the authors could mention the additional contribution of their manuscript to the existing literature. –

Thank you for the very constructive suggestion. All changes have been made in the revised manuscript, in red. We added a paragraph regarding the importance of extramedullary hematopoiesis in the diagnosis of hepato-splenomegaly.

Specific comments:

a lot of abbreviations are used in the manuscript, some of them are not commonly used, which makes the reading somehow confusing. I would suggest to reduce those abbreviations that are not necessary; also, for example hemoglobin is more commonly abbreviated as "HbA, HbF" instead of "HgA, HgF",  I therefore suggest to adopt commonly used abbreviations for a easier reading.

Thank you for the very constructive suggestion. All changes have been made in the revised manuscript, in red. We corrected the abbreviations and eliminated them in places that are not necessary.

a table is shown in the manuscript, but it is not clear its role and contribution to the manuscript. also, this table is not clearly mentioned in the text.

Thank you for the very constructive suggestion. All changes have been made in the revised manuscript, in red. We added the explanation of the table in the text.

line 106 : MPP to be defined

Thank you for the very constructive suggestion. We defined MPP

paragraph lines 142 to 150 the link between the different sentences of this paragraph is not clear , please revise.

Thank you for the very constructive suggestion. We revised the paragraph. All changes have been made in the revised manuscript, in red.

line 159: HSC are again defined but this is done since the beginning of the manuscript.

Thank you for the very constructive suggestion. We deleted the definition and kept only the abbreviation.

Table: line "Inflammation": in the "number of cases" column it is noted "in vitro, in vivo", please revise/complete.

Thank you for the very constructive suggestion. We revised the row by replacing "in vitro, in vivo" by N/A

table titer and number are missing.

Thank you for the very constructive suggestion. We added a table number and title. All changes have been made in the revised manuscript, in red.

Reviewer 2 Report

Extramedullary hematopoiesis with emphasis on MPNs is well described by Cenariu et al, however the clinical implications of hepatosplenomegaly and important differential diagnosis are not well described, also references are mainly not according to the journal style.

Author Response

Extramedullary hematopoiesis with emphasis on MPNs is well described by Cenariu et al, however the clinical implications of hepatosplenomegaly .

Thank you for the very constructive suggestion. We have made all the required changes in red, in the revised manuscript. We added a few lines about the clinical implications of hepato-splenomegaly and important differential diagnosis are not well described. We added a section discussing differential diagnosis of different malignant hemopathies, also references are mainly not according to the journal style.

Thank you for the very constructive suggestion. We formatted the bibliography according to MDPI style.
